# Mitochondrial Dysfunction Plays a Relevant Role in Pathophysiology of Peritoneal Membrane Damage Induced by Peritoneal Dialysis

**DOI:** 10.3390/antiox10030447

**Published:** 2021-03-13

**Authors:** Olalla Ramil-Gómez, Ana Rodríguez-Carmona, Jennifer Adriana Fernández-Rodríguez, Miguel Pérez-Fontán, Tamara Ferreiro-Hermida, Mirian López-Pardo, Teresa Pérez-López, María J. López-Armada

**Affiliations:** 1Aging and Inflammation Research Laboratory, Institute for Biomedical Research of A Coruña (INIBIC), 15006 A Coruña, Spain; Olalla.Ramil.Gomez@sergas.es (O.R.-G.); Jennifer.Fernandez.Rodriguez@sergas.es (J.A.F.-R.); mirian.lpardo@udc.es (M.L.-P.); 2Division of Nephrology, University Hospital A Coruña, 15006 A Coruña, Spain; arodtor@sergas.es (A.R.-C.); Miguel.Perez.Fontan@sergas.es (M.P.-F.); Tamara.Ferreiro.Hermida@sergas.es (T.F.-H.); Teresa.Perez.Lopez@sergas.es (T.P.-L.)

**Keywords:** peritoneal dialysis, epithelial-to-mesenchymal transition, oxidative stress, mitochondria, mitochondrial DNA, biomarker, mesothelial cells

## Abstract

Preservation of the peritoneal membrane is an essential determinant of the long-term outcome of peritoneal dialysis (PD). Epithelial-to-mesenchymal transition (EMT) plays a central role in the pathogenesis of PD-related peritoneal membrane injury. We hypothesized that mitochondria may be implicated in the mechanisms that initiate and sustain peritoneal membrane damage in this setting. Hence, we carried out ex vivo studies of effluent-derived human mesothelial cells, which disclosed a significant increase in mitochondrial reactive oxygen species (mtROS) production and a loss of mitochondrial membrane potential in mesothelial cells with a fibroblast phenotype, compared to those preserving an epithelial morphology. In addition, in vitro studies of omentum-derived mesothelial cells identified mtROS as mediators of the EMT process as mitoTEMPO, a selective mtROS scavenger, reduced fibronectin protein expression induced by TGF-ß1. Moreover, we quantified mitochondrial DNA (mtDNA) levels in the supernatant of effluent PD solutions, disclosing a direct correlation with small solute transport characteristics (as estimated from the ratio dialysate/plasma of creatinine at 240 min), and an inverse correlation with peritoneal ultrafiltration. These results suggest that mitochondria are involved in the EMT that human peritoneal mesothelial cells suffer in the course of PD therapy. The level of mtDNA in the effluent dialysate of PD patients could perform as a biomarker of PD-induced damage to the peritoneal membrane.

## 1. Introduction

Chronic kidney disease is a severe disorder that is reaching epidemic proportions [1]. Peritoneal dialysis (PD) is a growing therapeutic alternative for end-stage kidney disease [2]. In this technique, the peritoneum of the patient is used as a semipermeable membrane for the exchange of water and solutes between the blood and different solutions introduced (and periodically renewed) in the peritoneal cavity [3]. Unfortunately, long-term exposure to these non-physiologic PD solutions, as well as to the peritoneal catheter, may lead to progressive peritoneal membrane injury resulting in PD technique failure and even mortality [4]. Mesothelial cells lining the peritoneal cavity represent the first line of contact with PD fluids, and appear to play an active role in the loss of integrity of the peritoneal membrane during PD therapy [5,6,7,8].

In response to PD, mesothelial cells can react by losing their epithelial phenotype, acquiring myofibroblast-like characteristics through an epithelial-to-mesenchymal transition (EMT) process [7]. This process induces early pathological changes in the peritoneal membrane [9]. In addition, ex vivo studies have shown that these morphologic alterations correlate with disorders of peritoneal membrane transport characteristics [10]. Several molecular signaling pathways have been described to induce EMT, of which TGF-ß1 is one of the most powerful [11]. During EMT, mesothelial cells experience a decrease in the expression of epithelial markers, including E-cadherin, enhancing the expression of mesenchymal markers like fibronectin, collagen I or α-smooth muscle actin (α-SMA) [7]. As a consequence, cells acquire invasive capacities and reach the submesothelial stroma, where they produce extracellular matrix—but also inflammatory and angiogenic—factors, promoting peritoneal oxidative stress, inflammation and, finally, fibrosis, affecting peritoneal transport of water and solutes and resulting in ultrafiltration failure [6,12].

A large body of evidence supports the important role of oxidative stress in the peritoneal pathophysiology of patients undergoing PD. In fact, the presence of free radicals in the effluent is associated with technique failure in stable PD patients [13,14]. Previous studies have demonstrated that glucose-rich and/or acidic hypertonic PD solutions could be associated with higher levels of oxidative stress [15,16,17,18]. The PD-related oxidative process can be modulated by drug treatments used in these patients [19,20].

Mitochondria play a key role in oxidative stress since they represent the most important source of reactive oxygen species (ROS), but they are also targeted by these molecules [21]. Besides, damaged mitochondria release damage-associated molecular patterns (DAMPs, also known as alarmins), such as mtROS or mtDNA, which are recognized by the immune system and subsequently trigger an inflammatory and immune response, which can also cause mitochondrial damage, thus perpetuating a vicious cycle of mitochondrial alteration and activation of pathologic pathways [22].

Regarding PD, mitochondrial dysfunction is one of the main causes of glucose-rich-dialysate-induced mtROS and apoptosis, as well as mtDNA damage in human peritoneal mesothelial cells [23,24,25]. Plasma and dialysate mtDNA levels could perform as prognostic markers for peritoneal membrane function and PD technique survival [26,27]. All these considerations point to a potential involvement of mitochondrial dysfunction in the structural and functional disorders appearing in the course of PD therapy, but the role of mitochondria in the pathophysiology of PD-related peritoneal membrane damage has not yet been adequately defined.

The purpose of the present study was to investigate the role of mitochondrial dysfunction in the damage that the peritoneal membrane suffers during the course of PD. Secondarily, we explored the role of mtDNA as a potential biomarker of PD-induced damage to the peritoneal membrane.

## 2. Materials and Methods

### 2.1. Study Population

The study was conducted according to the Spanish Law for Biomedical Research (Law 14/2007-3 of July), complied with the Helsinki declaration and its later amendments and was approved by the local Ethics Committee (Galicia, Spain) (code number 2014/454). Written informed consent was obtained from all patients included in the study, and samples were coded to maintain anonymity. Demographic and clinical data were collected by medical record review.

### 2.2. Culture and Cell Stimulation of Human Mesothelial Cells

Human mesothelial cells were obtained from PD effluents and from omentum samples using standard methods described previously [10]. For ex vivo analysis, we collected peritoneal effluents from 118 patients treated with PD in the University Hospital of A Coruña. Dialysate samples were obtained at the time of the initiation of PD therapy (new initiation) or after overnight dwells, in the latter case simultaneously with routine performance of peritoneal equilibration tests (PETs) with complete drainage at 60 min. All patients were treated with low-glucose degradation product, bicarbonate-lactate buffered PD solutions. Samples were categorized as new initiation (PD time = 0 months), short vintage (PD time < 4 months) and long vintage (PD time ≥ 4 months). After centrifugation at 1500 rpm for 10 min, the cell pellet was cultured in Roswell Park Memorial Institute (RPMI) medium (Lonza, Basel, Switzerland) supplemented with 20% heat-inactivated fetal bovine serum (FBS; ThermoFisher, Hillsboro, OR, USA), 100 U/mL penicillin, 100 µg/mL streptomycin (Gibco, Hillsboro, OR, USA) and 0.12 U/mL human insulin (Novo Nordisk Pharma, Madrid, Spain) until reaching the confluence. Mesothelial cells were characterized by phase contrast microscopy (Nikon Eclipse TS100, Melville, NY, USA) according to their EMT status as epithelial or fibroblast-like, as previously reported [7]. In relation to membrane transport type, patients were classified as high, high-average, low-average and low according to the levels of 4-h dialysate/plasma (D/P) creatinine at ≥81, 66–80, 51–65 and ≤50, respectively.

For in vitro studies, human omental mesothelial cells were obtained from omentum from three non-PD patients who underwent unrelated elective abdominal surgery. The tissue was washed in saline serum solution (Fresenious, Barcelona, Spain) before being cut into pieces of about 4 mm^2^ and digested with trypsin-0.05% EDTA (ThermoFisher) for 20 min at 37 °C with agitation. The resulting suspension was centrifuged at 1200 rpm for 8 min and cells were grown in RPMI medium supplemented with 20% inactivated FBS, 100 U/mL penicillin, 100 µg/mL streptomycin and 0.12 UI/mL human insulin. TGF-β (1 ng/mL) (Abcam, Cambridge, UK) was used to induce EMT. The mitochondrial antioxidant mitoTEMPO (10 µM) (Santa Cruz Biotechnology, Santa Cruz, CA, USA) was used to scavenge mtROS.

### 2.3. Western Blotting Assay

Western blot analyses were performed utilizing standard procedures. Briefly, 35 μg of isolated protein from cell lysates were loaded and resolved using standard 7.5 % SDS-polyacrylamide gel electrophoresis (SDS-PAGE). The separated proteins were then transferred to polyvinylidene difluoride (PVDF) membranes (Immobilon P, Millipore, Billerica, MA, USA) by electroblotting, which were blocked with fat-free milk and probed with specific antibodies against E-cadherin (1:1000, BD Biosciences, San Diego, CA, USA) and fibronectin (1:5000, abcam). Immunoreactive bands were detected by chemiluminescence using corresponding horseradish peroxidase (HRP)-conjugated secondary antibodies and enhanced chemiluminescence (ECL) detection reagents (GE Healthcare, Bloomington, IL, USA), and then digitized using the LAS 3000 image analyzer. Tubulin (1:5000, Sigma, St. Louise, MO, USA) was used as loading control. Quantitative changes in band intensities were evaluated using Image J software (version 1.50; National Institute of Health, Maryland, WA, USA).

### 2.4. Mitochondrial ROS Detection

Mitochondrial ROS production was measured in mesothelial cells using MitoSOX Red (Life Technologies, Hillsboro, OR, USA). The dye permeates the membrane and become oxidized by superoxide exhibiting red fluorescence. Mesothelial cells from PD effluents were seeded in 48-well plates until confluence, washed with PBS and loaded with MitoSOX Red (0.8 ng/mL) in 200 µL of Hank’s Balanced Salt Solution (HBSS, Gibco, Hillsboro, OR, USA) for 30 min at 37 °C. Cells were subsequently washed, trypsinized and centrifuged and then fluorescence was measured by flow cytometry using a FACScalibur cytometer. The analysis of the data was performed using CellQuest Pro 5.1 software.

### 2.5. Assessment of Mitochondrial Membrane Potential

Mitochondrial membrane potential was determined using the fluorescent dye tetramethylrhodamine (TMRM, 250 nM) (Life Technologies). Polarized mitochondria accumulate more fluorescent dye, whereas depolarized mitochondria (lower mitochondrial membrane potential) retain less dye and, therefore, show lower fluorescence intensity. The fluorescence intensity was measured by flow cytometry using a FACScalibur cytometer (BD) and the data obtained were analyzed by CellQuest Pro 5.1 software (BD).

### 2.6. Dialysate mtDNA Level

Effluent samples from PD patients were centrifuged at 1500 rpm for 10 min, and 5 mL of supernatant was saved and stored at −80 °C immediately. DNA was extracted from 1 mL of supernatant using Circulating DNA Minikit (Danagen, Barcelona, Spain) according to the manufacturer’s instructions. The level of mtDNA was performed by real-time qPCR for the human mitochondrial 12S gene expression (forward 5′-CCACGGGAAACAGCAGTGAT-3′, reverse 5′-CTATTGACTTGGGTTAATCGTGTGA-3′) (Sigma). The LightCycler 480 (Roche, Indianapolis, IN, USA) system was employed with the following cycling conditions: denaturation step at 95 °C for 10 min, followed by 50 cycles at 95 °C for 10 s, 62 °C for 60 s and 72 °C for 5 s. The level of mtDNA copies was extrapolated into an external standard curve developed from serial dilutions of mtDNA isolated from Tc28a2 cells and was expressed as the mtDNA copy number/µL.

### 2.7. Statistical Analysis

Data were expressed as the mean ± SEM and a *p*-value of ≤0.05 was considered significant. Ex vivo statistical analysis (mtROS, mitochondrial membrane potential and mtDNA) was carried out using the SPSS 26 software (IBM, Manhattan, NY, USA). As the data were not normally distributed, non-parametric tests were carried out. To compare numerical values between two independent groups, the Mann–Whitney test was used; however, to compare differences between more than two independent groups, the Kruskall–Wallis test was performed. In the case of categorical data, the Chi-square test was applied. Finally, correlations between numerical variables were analyzed with Spearman’s test. The in vitro analysis was performed with GraphPad PRISM 7 software (GraphPad Software, San Diego, CA, USA) using the non-parametric Mann–Whitney test.

## 3. Results

### 3.1. Mitochondrial Dysfunction during the Epithelial-to-Mesenchymal Transition of Mesothelial Cells in PD Patients Ex Vivo

We tested the hypothesis that mitochondrial dysfunction could be involved in the epithelial-to-mesenchymal transition of human peritoneal mesothelial cells, a key process in peritoneal membrane failure. Firstly, effluent-derived mesothelial cells from 67 PD patients were classified as epithelial and non-epithelial according to their phenotype at confluence (Figure 1A). Secondly, as seen in Figure 1B,C, protein expression studies of epithelial marker E-cadherin and mesenchymal marker fibronectin were carried out, showing that cells with epithelial phenotype had a positive expression of E-cadherin, which was almost completely absent in non-epithelial cells. By contrast, fibronectin protein expression in epithelial cells was scant, and significantly increased in cells with a non-epithelial phenotype (Figure 1B,C). The baseline characteristics of the patients according to the morphology of effluent-derived mesothelial cells are shown in Table 1. Thirdly, quantitative studies performed by flow cytometry showed that effluent-derived mesothelial cells with a non-epithelial phenotype produced a significantly higher level (*p* ≤ 0.05) of mtROS than the epithelial ones (Figure 2A). Likewise, a significant decrease (*p* ≤ 0.05) of mitochondrial membrane potential was observed in cells that suffered the EMT process (Figure 2B) as assessed by decreased tetramethylrhodamine methyl ester staining. Furthermore, when effluent-derived mesothelial cells were classified according to PD duration in new initiation (PD time = 0 months) and PETs (PD time > 0 months), an increasing trend in the level of mtROS (new initiation: 6.6 ± 0.6 vs. PETs: 9.0 ± 0.7) and a decrease of mitochondrial membrane potential (new initiation: 280.4 ± 31.0 vs. PETs: 234.9 ± 10.4) were observed in mesothelial cells derived from PET samples, but they did not reach statistical significance when compared with mesothelial cells derived from new initiation patients.

### 3.2. Mitochondrial Dysfunction during the Epithelial-to-Mesenchymal Transition of Mesothelial Cells

The mitochondrial alterations during EMT were confirmed in vitro using TGF-β1, the key molecule in the induction of EMT. As seen in Figure 3, fibronectin protein expression, induced by TGF-β1 (1 ng/mL), was significantly reversed (*p* ≤ 0.05) by inhibiting mtROS production when human mesothelial cells from omentum samples were first preincubated with the mitochondria-targeted antioxidant mitoTEMPO (10 µM). Therefore, these results further confirm the involvement of the mitochondria in the EMT process, since a reduction in ROS production by this organelle is translated into a decrease in fibrosis.

### 3.3. Free mtDNA Is Highly Elevated in the Peritoneal Effluent from PD Patients’ New Initiation

Next, we measured mtDNA concentration in the dialysate fluid in 232 samples from 118 patients. Table 2 shows the demographic and clinical characteristics of patients classified according to time in PD in new initiation, short vintage and long vintage samples. Surprisingly, as seen in Figure 4, the results obtained indicated that there was a significant increase of mtDNA copy number in the effluent of the new initiation samples (1147 ± 334), in relation to a longer duration of PD, compared to PET samples, both short (71 ± 22, *p* ≤ 0.001) and long vintage (47 ± 10, *p* ≤ 0.0001). Relatedly, there was a negative and significant correlation between free mtDNA levels and time in PD (Spearman’s Rho = −0.367, *p* ≤ 0.001, *n* = 227). The differences observed in the mtDNA levels between short and long vintage patients were not statistically significant (*p* = 0.63). No significant differences were found between surgical and percutaneous catheter implants. Also, no correlation was found between plasma PCR levels and dialysate free mtDNA levels (Spearman’s Rho = −0.008, *p* = 0.913).

### 3.4. Correlation Between free mtDNA in the Peritoneal Dialysate Effluent from Long Vintage Patients and the Peritoneal Transport Rate

As a final step, we explored the potential relationship between mtDNA levels in spent dialysate and peritoneal transport characteristics, estimated from the D/P creatinine ratio and standardized ultrafiltration during the PET studies. To avoid interference from the initial damage to the peritoneal membrane caused by catheter insertion, and considering that PET studies performed in long vintage patients may be more representative of the long-term peritoneal transport characteristics of the patients, this analysis was carried out only in long vintage patient samples (time on PD > 4 months, 37.7 ± 29.4 (SD) months, *n* = 113). As shown in Figure 5A, our results disclosed a positive correlation between dialysate mtDNA copy number and the D/P creatinine ratio in long vintage patients (Spearman’s Rho = 0.309, *p* ≤ 0.001, *n* = 112), although comparisons between different patient subsets, categorized according to membrane transport characteristics and mtDNA levels, showed marked differences amongst patients within the same transport type (Figure 5B). Furthermore, we observed a negative correlation between dialysate mtDNA copy numbers and ultrafiltration rates in long vintage patients (Spearman’s Rho = −0.251, *p* ≤ 0.01, *n* = 112).

## 4. Discussion

There is a clear connection between mitochondrial damage, oxidative stress and inflammation [21,28], as well as between chronic inflammation and fibrosis, including peritoneal fibrosis [10,29]. According to this point of view, inflammation could antedate membrane fibrosis, and even the relationship between these two processes could be bidirectional, each one inducing the other [30,31]. In the last few years, several studies have related mitochondrial impairment with peritoneal membrane damage in the course of PD therapy [23,25]. However, the possible role of mitochondrial dysfunction in the EMT process that mesothelial cells from peritoneal membrane suffer in patients treated with PD, as well as in monitoring PD-induced damage to the peritoneal membrane, remains unknown.

Some of our present findings have not, to our knowledge, been reported so far. Mesothelial cells play a relevant role in the structural and functional alterations affecting the peritoneal membrane in the course of PD [7]. PD fluids cause by themselves the EMT of mesothelial cells both in vivo and in vitro [32]. This is the first study to document mtROS production and mitochondrial membrane potential in cultured human mesothelial cells derived from PD effluent. On one hand, our ex vivo model shows that effluent mesothelial cells with a non-epithelial phenotype undergo an increased production of mtROS when compared to those displaying an epithelial phenotype. This finding suggests that mesothelial cells, which have undergone EMT, are an important source of mtROS, and could contribute to the peritoneal inflammatory response and, as a consequence, to progression of membrane fibrosis, which is the main feature of peritoneal membrane injury during PD [7,9]. In fact, when omentum-derived mesothelial cells were treated with the mitochondria-targeted antioxidant mitoTEMPO, the protein fibronectin expression induced by TGF-β1 was significantly decreased. These findings confirm ROS, and specifically mtROS, as key mediators in the process of PD-mediated EMT. These results are consistent with in vitro studies in other cell types, and also in mesothelial cells, showing that TGF-β1 also increases the generation of mtROS, resulting in inflammatory response, phenotype transition and, finally, fibrosis [19,20,29,33]. On the other hand, we also observed a significant decrease of mitochondrial membrane potential in cells affected by EMT. Overall, our results could indicate that mesothelial cells of PD patients suffer mitochondrial damage during PD treatment, causing an increase of mtROS and a depolarization of the mitochondrial membrane. Remarkably, decline of mitochondrial membrane potential is a feature of apoptotic cell death [34]. These results are consistent with previous studies describing apoptotic mesothelial cells in the peritoneal effluent of PD patients and dialysate-glucose-induced oxidative stress and mitochondrial-mediated apoptosis in human peritoneal mesothelial cells [23,25]. Moreover, biocompatible PD fluids may reduce apoptosis of mesothelial cells [24]. Other studies on lung epithelial cells reported that reduction in mitochondrial respiration could lead to increased ROS and decreased mitochondrial membrane potential [35]. A more recent study has described how inhibition of hyperglycolysis in mesothelial cells prevents peritoneal fibrosis by suppressing EMT [12]. In this sense, TGF-β1 can be substituted for PD fluid to stimulate hyperglycolysis, suppressing mitochondrial respiration in mesothelial cells, as well as dynamic mitochondrial changes [12]. These findings underline the relevance of our findings with regard to the effects of mitochondrial dysfunction.

The next set of experiments focused on the analysis of mtDNA concentration in the PD effluent. Mitochondria have been recently identified as key sources of DAMPs (mito-DAMPs), playing a crucial role in DAMP-modulated inflammatory and immune responses [22,36]. Specifically, mtDNA is a marker of mitochondrial damage, and its presence is indicative of mitochondrial lysis. mtDNA can be released following different types of cell death, such as apoptosis [37]. mtDNA has been implicated in NLRP3 inflammasome activation, a cytosolic receptor that once activated induces the maturation of the proinflammatory cytokines IL-1β and IL-18 [38]; and also in Toll-like membrane receptors (TLRs) activation, resulting in an inflammatory response [39]. Interestingly, the activation of TLR9 by mtDNA mediates TGF-β1-induced fibroblast activation in lung tissue, potentially leading to fibrosis [40]. Regarding PD, plasma and dialysate mtDNA levels have been pointed out as prognostic markers of PD patient survival and peritoneal solute transport in short vintage patients, respectively [26,27]. Also, NLRP3 is involved in PD-related peritonitis [38].

Considering that local intraperitoneal inflammation is an important determinant of the peritoneal solute transport rate [41] and that the kinetics of peritoneal transport are more representative of long-term peritoneal function in long than in short vintage patients [42], our next step was to analyze in the former subset the relationship between dialysate mtDNA concentration and peritoneal membrane transport characteristics. Our data disclosed that stepwise increments in effluent mtDNA levels were associated with faster solute transport rates and a decreasing capacity of ultrafiltration. The previous study from Xie et al. also described dialysate cell-free mtDNA fragments as a marker of peritoneal solute transport rate in peritoneal dialysis [26]. However, the median time of PETs in this study was only 2.7 months after initiation of PD (taking in consideration samples from the first PET analysis to 6 months of treatment). In contrast, the median time of PET in our correlation study was 37.7 months after initiation of PD (PD time ≥ 4 months), which allowed us to evaluate this parameter in the long term. Even though there is a statistically significant direct correlation between the peritoneal solute transport rate and free mtDNA levels in the PD effluent, the latter showed a marked variability amongst patients with similar transport rates, suggesting that other factors, such as genetic predisposition or diet, could influence the background properties of the peritoneal membrane [43,44,45]. In support of our findings, it has been shown that mtDNA increases enhance endothelial permeability through different pathways [46]. Additionally, our results suggest a role for mito-DAMPs as important therapeutic targets in conditions where inflammation increases pathologically endothelial permeability, as in the case of PD [46]. Our findings are also consistent with those obtained in idiopathic pulmonary fibrosis bronchoalveolar lavage samples, which exhibit a high concentration of extracellular mtDNA [40]. Also, idiopathic pulmonary fibrosis fibroblasts exhibit high concentrations of extracellular mtDNA, which is able to induce fibrosis in normal human lung fibroblasts [40]. Additionally, dialysate IL-6 concentration, representing local subclinical intraperitoneal inflammation, is a well-known predictor of the small solute transport rate in PD [41]. Previous studies have shown how dialysate mtDNA levels are significantly associated with IL-6 levels in short-vintage PD patients [26]. Preliminary results from our group showed an increase in effluent IL-6 and mtDNA levels in patients whose effluent-derived mesothelial cells were non-epithelioid phenotype (unpublished data), although the differences did not reach statistical significance. Likewise, a recent in vitro model shows how mito-DAMPs-treated peripheral blood mononuclear cells secreted IL-6 that impaired mitochondrial respiration [47]. In this regard, more studies are needed to understand the association between local inflammation, the EMT process and peritoneal transport characteristics in the long term.

Our data disclosed markedly increased mtDNA copy numbers in the effluent of the new initiation, as compared with samples obtained later in the course of PD. In agreement, there was a significant inverse correlation between free mtDNA levels and time on PD. No significant differences were found between short and long vintage patients. This increase in mtDNA levels at the very inception of PD could reflect an acute response of the peritoneum to tissue injury caused by the catheter insertion. This increase in the level of free mtDNA could be a consequence of leukocyte rolling and extravasation by peritoneal catheter insertion, acting as a foreign body [48]. In addition, there was no difference in mtDNA levels between percutaneous and surgical placement (data not shown), although, in this particular case, our analysis was hampered by a limited statistical power. On the other hand, we were not able to detect an association between dialysate free mtDNA and plasma C-reactive protein levels. This circumstance is not unexpected, as previous studies have established the independence of plasma and dialysate inflammatory markers in PD patients [41]. Alternatively, it is known that peritoneal cavity lavage reduces the presence of mito-DAMPs in open abdomen patients, and it is possible that frequent peritoneal cavity lavage during PD may lead to decreased systemic absorption of mito-DAMPs, thereby reducing the risk of systemic inflammatory response syndrome [49]. It is also possible that recurrent drainage of dialysate could be the reason why the high levels of mtDNA that were observed in the new initiation did not reach the bloodstream and, consequently, why the inflammatory response was reduced at the systemic level. The potential consequences of the increased free mtDNA released at inception of PD are unclear, but could be relevant, since mtDNA is able to trigger an inflammatory response followed by fibrosis and, lastly, loss of integrity of the peritoneal membrane.

Our results provide evidence that mitochondrial dysfunction promotes EMT in mesothelial cells of PD patients. Conversely, preserving mitochondrial activity could suppress mesothelial EMT and peritoneal fibrosis. In this regard, it has recently been showed how metformin, an inductor of autophagy, could prevent mtDNA release and inhibit EMT and peritoneal fibrosis via a modulation of oxidative stress [19]. In addition, the level of mtDNA in the dialysate could help to monitor PD-induced damage to the peritoneal membrane, acting as an indicator of the dialytic capacity, as well as a marker of the intraperitoneal inflammation. Overall, mitochondrial function preservation could help to limit peritoneal membrane injury during PD, as well as EMT and fibrogenesis in other organs. Clearly, more studies are required to elucidate the role of mitochondrial dysfunction in the pathophysiology of peritoneum in PD patients.

## Figures and Tables

**Figure 1 antioxidants-10-00447-f001:**
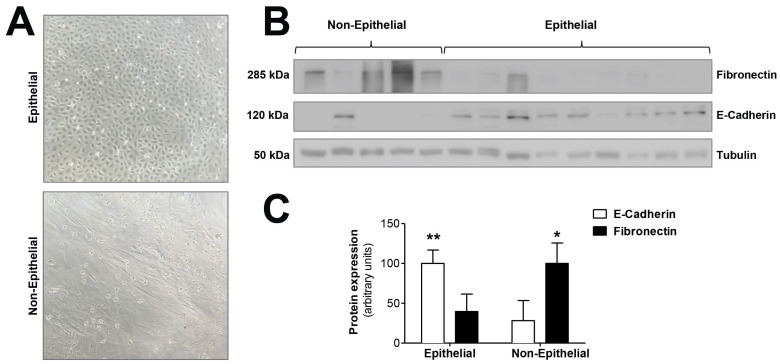
Epithelial-to-mesenchymal transition of mesothelial cells during PD. (**A**) Phase-contrast microscopy showing different morphologic characteristics of mesothelial cells with epithelial and non-epithelial phenotypes. Original magnification ×10. (**B**) Western blot showing the expression of typical markers of epithelial-to-mesenchymal transition, E-cadherin (epithelial marker) and fibronectin (mesenchymal marker). (**C**) The graph shows the mean ± SEM levels of expression in nine epithelial samples and five non-epithelial samples; ** *p* < 0.01 and * *p* < 0.05 vs. non-epithelial and epithelial cells, respectively. PD: peritoneal dialysis.

**Figure 2 antioxidants-10-00447-f002:**
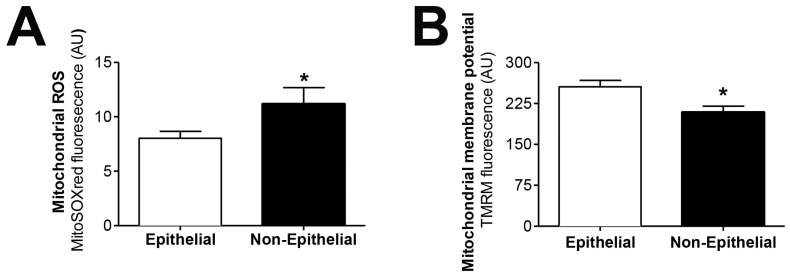
Effluent-derived human mesothelial cells with non-epithelial phenotype exhibit increased mtROS and decreased mitochondrial membrane potential. (**A**) Epithelial and non-epithelial mesothelial cells were stained with MitoSOX Red and analyzed by flow cytometry. Values are the mean ± SEM median fluorescence intensity (*n* = 62 epithelial samples and *n* = 17 non-epithelial samples). * *p* < 0.05 vs. epithelial cells. (**B**) Epithelial and non-epithelial mesothelial cells were loaded with tetramethylrhodamine methyl ester and analyzed by flow cytometry. Values are the mean ± SEM median fluorescence intensity (*n* = 62 epithelial samples and *n* = 15 non-epithelial samples). * *p* < 0.05 vs. epithelial cells. PD: peritoneal dialysis.

**Figure 3 antioxidants-10-00447-f003:**
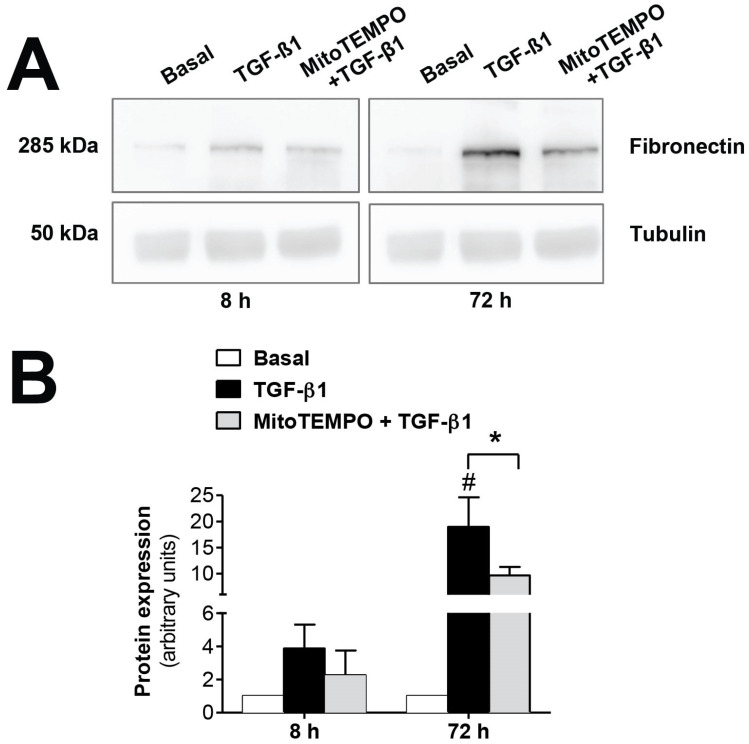
Modulatory effects of MitoTEMPO on TGF-β1-induced epithelial-to-mesenchymal transition (EMT) in omentum-derived mesothelial cells. (**A**) Representative Western blot analysis of fibronectin expression at 8 and 72 h in TGF-β1-treated mesothelial cells with and without Mito TEMPO pretreatment and (**B**) quantification expressed in arbitrary units. Bars represent the fold induction over untreated cells as means ± SEM of three independent experiments. * *p* < 0.05 vs. TGF-β1 and # *p* < 0.05 vs. Basal. PD: peritoneal dialysis; EMT: epithelial-to-mesenchymal transition.

**Figure 4 antioxidants-10-00447-f004:**
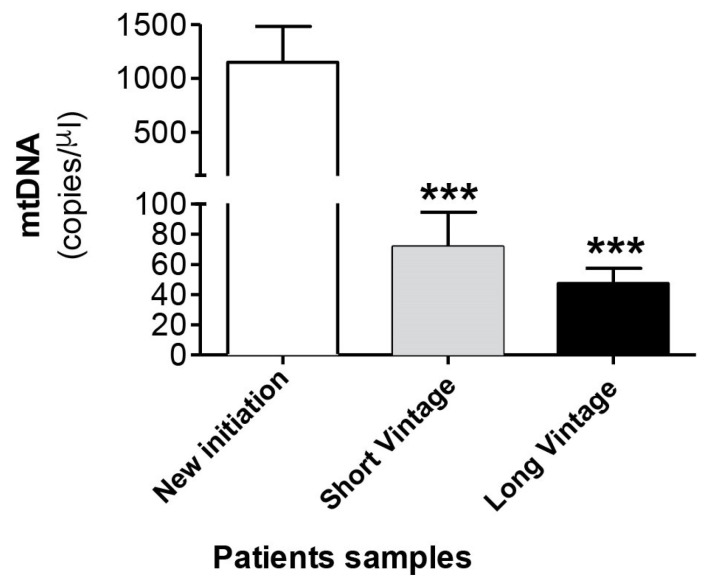
mtDNA levels in the peritoneal effluent from PD patients. Values are the mean ± SEM mtDNA copy numbers of patients classified according to time in PD in new initiation (*n* = 49), short vintage (*n* = 70) and long vintage (*n* = 113) samples. *** *p* < 0.001 vs. first connection. PD: peritoneal dialysis.

**Figure 5 antioxidants-10-00447-f005:**
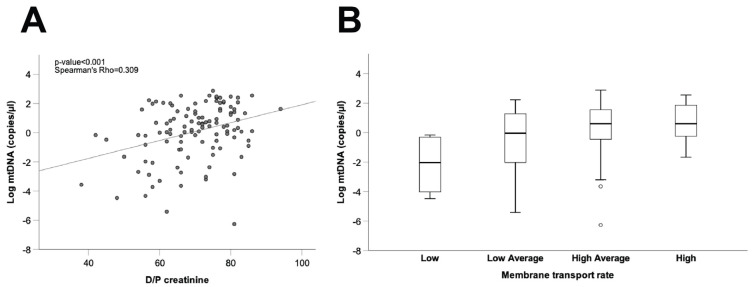
Correlation between free mtDNA in the PD effluent from long vintage patients and the peritoneal transport rate. (**A**) Correlation between mtDNA copy number and D/P creatinine. Spearman’s Rho and *p*-values are indicated (*n* = 113). (**B**) Comparison of PD effluent mtDNA levels between patient groups according to membrane transport type. Data are presented graphically as log base 10 of the raw values of mitochondrial 12S copies per µL of PD effluent. PD: peritoneal dialysis; D/P creatinine: dialysate/plasma creatinine ratio.

**Table 1 antioxidants-10-00447-t001:** Study population characteristics according to the morphology of effluent-derived mesothelial cells.

Variable	Total	Epithelial	Non-Epithelial	*p*-Value
Number of samples	80	63	17	-
Number of patients	67	-	-	-
Age (years)	62.7 ± 1.7	62.0 ± 2.0	65.3 ± 3.6	0.466
Body mass index (Kg/m^2^)	27.6 ± 0.6	27.5 ± 0.7	28.1 ± 1.3	0.638
Male (%)	61.2	61.1	61.5	0.977
Diabetic (%)	38.8	37	46.2	0.545
Time on PD (months)	16.7 ± 2.6	14.2 ± 2.8	26.1 ± 5.7	0.006
New initiation/PET samples (%)	15.8/84.2	20.3/79.7	0/100	0.058
Short/long vintage patients (%)	39.1/60.9	44.7/55.3	23.5/76.5	0.126
Glomerular filtration rate (mL/min)	6.2 ± 0.4	6.2 ± 0.5	6.5 ± 0.9	0.769
D/P Creatinine at 4 h	72.5 ± 1.1	72.1 ± 1.2	73.4 ± 2.7	0.484
D/D_0_ glucose at 4 h	28.5 ± 1.1	29.1 ± 1.1	26.9 ± 2.7	0.250
Na sieving (mM/L)	8.0 ± 0.6	7.9 ± 0.7	8.3 ± 1.1	0.632
Ultrafiltration (mL)	391.4 ± 29.0	409.0 ± 32.0	342.6 ± 63.9	0.243
Peritoneal transport status (%):				
o Fast	14.1	8.5	29.4	-
o Fast-average	64.1	72.3	41.2	-
o Slow-average	21.9	19.1	29.4	-
o Slow	0	0	0	-
C-reactive protein (mg/dL)	0.7 ± 0.1	0.8 ± 0.1	0.4 ± 0.1	0.382

PD: peritoneal dialysis; PET: peritoneal equilibration test; D/P creatinine: dialysate/plasma creatinine ratio; D/D_0_ glucose: dialysate glucose at 4 h dwell time to dialysis glucose at 0 h dwell time ratio. Data are expressed as the mean ± SEM.

**Table 2 antioxidants-10-00447-t002:** Demographic and clinical characteristics of patients classified according to time on PD in new initiation, short vintage and long vintage samples.

Variable	Total	New Initiation	Short Vintage	Long Vintage	*p*-Value
Number of samples	232	49	70	113	-
Number of patients	118	49	30	39	-
Age (years)	63.3 ± 1.3	61.5 ± 2.0	65.4 ± 2.6	63.9 ± 2.2	0.535
Body mass index (Kg/m^2^)	28.3 ± 0.5	28.9 ± 0.8	27.4 ± 1.0	28.3 ± 0.8	0.456
Male (%)	59.8	59.2	58.6	61.5	0.074
Diabetic (%)	37.3	38.8	40	33.3	0.818
Time in PD (months)	19.6 ± 1.8	0.3 ± 0.3	2.7 ± 0.1	37.7 ± 2.8	0.000
Epithelial/Non ephithelial (%)	78.3/21.7	100/0	84.4/15.6	66.1/33.9	-
Glomerular filtration rate (mL/min)	5.8 ± 0.3	7.8 ± 1.1	7.8 ± 0.6	4.5 ± 0.3	0.000
D/P Creatinine 240 min	70.5 ± 0.8	-	71.4 ± 1.3	69.9 ± 1.0	0.351
D/D_0_ glucose 240 min	29.6 ± 0.7	-	28.0 ± 1.0	30.6 ± 1.0	0.184
Na sieving (mM/L)	8.0 ± 0.4	-	8.3 ± 0.6	7.8 ± 0.5	0.467
Ultrafiltration (mL)	431.6 ± 19.3	-	442.6 ± 33.1	424.7 ± 23.7	0.866
Peritoneal transport status (%)					
o Fast	13.7	-	18.6	10.7	-
o Fast-average	57.7	-	52.9	60.7	-
o Slow-average	25.8	-	27.1	25	-
o Slow	2.7	-	1.4	3.6	-
C-reactive proten (mg/dL)	0.7 ± 0.1	0.7 ± 0.2	0.9 ± 0.2	0.6 ± 0.1	0.724

PD: peritoneal dialysis; D/P creatinine: dialysate/plasma creatinine ratio; D/D_0_ glucose: dialysate glucose at 4 h dwell time to dialysis glucose at 0 h dwell time ratio. Data are expressed as the mean ± SEM.

## Data Availability

The data presented in this study are available on request from the corresponding author.

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
