# Peer review of "Mitochondrial Dysfunction Plays a Relevant Role in Pathophysiology of Peritoneal Membrane Damage Induced by Peritoneal Dialysis"

_antioxidants, 2021, doi:10.3390/antiox10030447_

Round 1
Reviewer 1 Report
This original manuscript by Ramil-Gomez el al. shows the roles of mitochondria in the peritoneal membrane physiology in a patient on peritoneal dialysis. This study was well designed experimental study.
In discussion, mtDNA has been implicated in inflammation (NLRP3 inflammasome, proinflammatory cytokines, TLR activation etc.). So, if possible, authors show the relationship between mtDNA and inflammation. It may be clearer in conclusion in role of mitochondrial dysfunction in peritoneal membrane characteristics.
Reviewer 2 Report
Authors collected mesothelial cells from PD effluent and analyzed ex vivo, for 118 patients with variable vintage of PD (month 0, less than 4 months, and longer then 4 months). The authors observed the changes in phenotypes of the mesothelial cells from epithelial to mesenchymal type, and the changes occur more frequently in patients with longer PD vintage. Additionally, they found EMT was associated with lower mtDNA. While the idea of EMT of peritoneal mesothelial cells and the association of mtDNA and membrane transport rate are not novel, the data of mitoROS and mito transmembrane potential provides some new insights.
Major:
- It seems the level of mtDNA is lower in the patients with longer PD vintage. Are they statistically significant? In Figure 4, the authors compared mtDNA values with the patients with “1st connectors” but didn’t compared between short (<4 mo) and long (>4 mo) vintages.
- It’d be interesting to measure cytokine levels of PD effluent, as the authors show TGF-b as a major driver of EMT. Would analyze other pro-inflammatory cytokines IL-6, IL-17, TNFa etc. (as mentioned in Xie et al. BMC Nephrology 2019)
- The reviewer wonders if the EMT is driven by the exposure of mesothelial cells to non-physiological PD fluids, or additional chronic inflammatory signals are needed. It’d be interesting to see if it is possible to observe EMT after long incubation in PD solutions.
Minor:
- The classification of patients with variable PD vintage with “first connection, incident, and prevalent” may be a misnomer: would suggest changing the name to “new initiation, short vintage of PD (<4 months) and long vintage of PD (>4 months)”.
- In figure 2B, please change the y-axis cut off. Using 200 AU cut off for y-axis skews the data interpretation.
- Typographical error: p.2. L48 experimentïƒ experience
Round 2
Reviewer 2 Report
The authors answered the suggestions and questions from the reviewer appropriately, and the manuscript has been significantly improved.